# Uses, Knowledge and Extinction Risk Faced by *Agave* Species in Mexico

**DOI:** 10.3390/plants12010124

**Published:** 2022-12-27

**Authors:** Cecilia Alducin-Martínez, Karen Y. Ruiz Mondragón, Ofelia Jiménez-Barrón, Erika Aguirre-Planter, Jaime Gasca-Pineda, Luis E. Eguiarte, Rodrigo A. Medellin

**Affiliations:** 1Escuela de Ciencias, Universidad de las Américas Puebla, Puebla 72810, Mexico; 2Instituto de Ecología, Universidad Nacional Autónoma de Mexico, Mexico City 04510, Mexico

**Keywords:** biodiversity, conservation, genetic diversity, distilled beverages, mezcal, pollinators

## Abstract

We compiled an updated database of all *Agave* species found in Mexico and analyzed it with specific criteria according to their biological parameters to evaluate the conservation and knowledge status of each species. Analyzing the present status of all *Agave* species not only provides crucial information for each species, but also helps determine which ones require special protection, especially those which are heavily used or cultivated for the production of distilled beverages. We conducted an extensive literature review search and compiled the conservation status of each species using mainstream criteria by IUCN. The information gaps in the database indicate a lack of knowledge and research regarding specific *Agave* species and it validates the need to conduct more studies on this genus. In total, 168 *Agave* species were included in our study, from which 89 are in the subgenus *Agave* and 79 in the subgenus *Littaea*. *Agave lurida* and *A. nizandensis*, in the subgenus *Agave* and *Littaea*, respectively, are severely endangered, due to their endemism, lack of knowledge about pollinators and floral visitors, and their endangered status according to the IUCN Red List. Some species are at risk due to the loss of genetic diversity resulting from production practices (i.e., *Agave tequilana*), and others because of excessive and unchecked overharvesting of wild plants, such as *A. guadalajarana*, *A. victoriae-reginae*, *A. kristenii*, and others. Given the huge economic and ecological importance of plants in the genus *Agave*, our review will be a milestone to ensure their future and continued provision of ecosystem services for humans, as well as encouraging further research in *Agave* species in an effort to enhance awareness of their conservation needs and sustainable use, and the implementation of eco-friendly practices in the species management.

## 1. Introduction

Mexico is well-known for its high levels of biodiversity, and agaves or magueyes belonging to the family Asparagaceae are a very important example of that biodiversity. The genus *Agave* is endemic to the American continent (although introduced *Agave* are now found in many places), mainly in arid and semi-arid areas of Mexico. There are approximately 200 species of *Agave*, of which 166 are distributed in Mexico and 119 are endemic [1,2]. The subgenus *Littaea* contains about 78 species (Figure 1) while the subgenus *Agave* includes around 92 species (Figure 2), making it the subgenus with the greatest distribution throughout Mexico [3]. Subgenera are distinguished by their large inflorescence, either a spike for the subgenus *Littaea* species or branched with panicles for *Agave* [4]. The succulence and rigidity of its leaves for water storing, the efficient water absorption system throughout its roots and the limited water loss through transpiration, are some of the adaptations agaves possess for life in dry ecosystems. Their CAM-type photosynthetic metabolism (Crassulaceae Acid Metabolism) allows them to grow in poor soil and weather conditions, withstanding severe environmental pressures [2]. These morphological and physiological adaptations have allowed agaves to live in a wide variety of environments, expanding their geographical distribution [3,5].

From a historical perspective, agaves have played a crucial role in the daily lives of human beings, providing a wide range of resources, from food and the production of alcoholic beverages to a source of fiber. The procurement of fibers from these plants was perhaps the first use given to agaves, since it was used for clothing, baskets, nets, rope, and more [4]. Several species have been used for food, the flowers boiled and eaten, and earthen ovens to cook agave cores or “piñas”. This method was an ancestral practice that is nowadays utilized for the production of distilled beverages [6]. The use of *Agave* to produce the distilled liquors known today as tequila, mezcal, and others was apparently not known to past civilizations, since distillation was introduced after the conquest, but the production of pulque and mead or “aguamiel” has been used since pre-Hispanic times, making these beverages the ones that promoters of the agave cultivation in Mexican civilizations. Currently over 70 traditional uses have been known and recorded for the species of the genus *Agave* [7], classified under 22 categories [8].

Additionally, the residues derived from the elaboration of distilled beverages, including the flowering stalks, bagasse (solid waste) and stillage (liquid waste) have various uses. The leaves, removed from the core of the agave rosette are commonly used in the preparation of Mexican dishes, such as in the “barbacoa” where the leaves are roasted and used to wrap the meat with spices for cooking while bagasse and stillage residues have been proven as a good alternative for compost [9]. These wastes are polysaccharide-rich byproducts that can be used for the production of fructans, which can be used in supplement products to treat diabetes or obesity [2]. 

*Agave* fibers have a wide variety of uses: as food for livestock and the production of different handcrafts, or common products such as nets, ropes or textiles. In rural areas, agaves are widely used as living fences to keep livestock away from crops or houses, to indicate roads, and even for construction [10]. *Agave sisalana* (sisal) and *A. fourcroydes* (henequen) are the most useful species for obtaining fiber and are widely used for structural construction material. *Agave sisalana*, for example, is still a major source of fiber in plantations in many countries of Latin America, Asia and Africa [11]. Fiber obtained from agaves is having a greater advantage over synthetic fibers nowadays as they are more affordable, with lower density, ecofriendly, and can be recycled [12].

Agaves are greatly known in the distilled beverages industry. Tequila and mezcal have become symbols of Mexico throughout the world, while pulque, the pre-Hispanic fermented alcoholic beverage, is considered to be one of the first traditional drinks originated from agave. Bacanora and raicilla are some of the least known agave distillates as they have a low level of industrialization compared to the rest of the distilled beverages [6,8].

Pulque is a white fermented beverage that can be obtained from a wide variety of agave species such as *A. atrovirens*, *A. americana*, *A. salmiana* and *A. mapisaga*. This drink is made through the fermentation of mead or *aguamiel*, a sugary extract that accumulates inside the core of the ripe agave and that can also be consumed fresh without fermenting [13]. Pulque is consumed both in rural areas, where it may even be considered as an important meal in a daily diet, and in urban areas, where it can be found in *pulquerias* and Mexican restaurants. Currently, the consumption of pulque is promoted partly so as not to lose a pre-Hispanic tradition [14,15].

Tequila is considered the most representative drink in Mexico, a job and currency generator of foreign exchange for its sales abroad. In 2016, Jalisco obtained $1.6 billion dollars in tequila exports abroad growing to $2.3 billion dollars in 2020 while the industry generated 70,000 jobs [16,17]. Tequila is obtained from *Agave tequilana* or blue agave, which can only be grown in regions known as Denomination of Origin of Tequila (DOT). This includes the states of Jalisco (126 municipalities) with 90% of production, Michoacán (30 municipalities), Tamaulipas (11 municipalities), Nayarit (eight municipalities) and Guanajuato (seven municipalities) [18]. Since *A. tequilana* is not common in the wild, tequila is sourced entirely from cultivated plants. In addition, to avoid a perceived waste if the plants are allowed to flower, agave fields are re-planted with young agave plants that are the result of asexual reproduction from a subset of the original plants in the field, thus resulting in a severe loss of genetic diversity generation after generation [19]. In addition, tequila has the most industrialized production processes of all distilled beverages originated from agave. The use of autoclaves (steam-fueled pressure-cookers) to cook the heads of *A. tequilana* is the most production-efficient method for *Agave* distillates, having control of the pressure-temperature ratio and its highly reduced cooking time compared to handcrafted methods [2,20,21]. Yet this process has serious costs for the quality and organoleptic characteristics of the final product as using diffusers for the thorough extraction of the sugars causes the addition of other secondary metabolites to the must, the fluid subjected to distillation, lowering the quality of the final product [22].

Mezcal, unlike tequila, is obtained from more than 50 agave species mostly found in the wild in Mexico. In 2007, Colunga-GarcíaMarín stated that at least 56 taxa are used for the production of this beverage. There is a wide diversity in mezcal production, which is due both to the diversity of the agave species used, and the variety of production processes. Thus, while tequila production is standardized and industrialized, the production of mezcal is generally handcrafted varying between regions so its flavor, aroma, alcohol content and quality depends on the region of production, the species of agave used and the mezcal manufacturer in charge of it [23]. Although mezcal production is generally handcrafted, industrial and so-called “ancestral” production processes are also used. Yet generally, traditional mezcal usually uses pit ovens or masonry pools for fermentation, grinding is carried out with tahona or Chilean mill, and distillation is made in copper stills or clay pots.

Bacanora is the local mezcal of the state of Sonora and is obtained from a variety of *A. angustifolia*, from which mezcal espadín is derived [24]. The variety *A. angustifolia* var. *pacifica* differs mainly in the cultivation of the agave as it is a wild agave with no use of fertilizers or pesticides and its distillation replaces the copper stills with metal barrels heated with mesquite wood. Currently, few producers are dedicated to the production and marketing of bacanora [25,26].

Coastal raicilla is obtained from *A. rhodacantha* and *A. angustifolia*, while the highland raicilla is acquired from *A. maximiliana*. This was a popular mezcal among miners of the mountains of the Sierra Madre Occidental of Jalisco and its production is currently only locally handcrafted by around 70 producers in 2014 [2,27].

The consumption of agave-derived products has increased in recent years and therefore its demand is rapidly growing nationally and internationally, mainly that of the distilled beverages. This demand implies a greater quantity of supply and modern technology in production processes, thus decreasing traditional methods and craftsmanship. That being said, it is important to emphasize that depending upon the *Agave* species and the local conditions of soil, humidity and nutrients, it may take between seven, ten, or 36 years or more (i.e., *A. tequilana*, *A. potatorum*, *A. seemanniana*, respectively) [4,28,29,30] for them to reach harvest maturity, so if sustainable practices and adequate environments are not secured for wild individuals, the future of their populations could be in jeopardy.

In the case of mezcal, this evolving tradition and increased demand has forced some producers to incorporate new technologies and infrastructures to be capable of adapting to a rapidly growing and demanding market, yet if proper management practices of agave populations are lost, it could result in a reduction of population size and genetic variability. Species such as *A. tequilana* for making tequila or *A. fourcroydes* to manufacture henequen, use clonal propagation, where suckers of the same genotype are grown and thus their genetic diversity is eroded generation after generation. The loss of genetic diversity carries a reduction of its adaptability to environmental and climate changes, pathogens and pests [2,23,30,31,32,33].

As with any species, genetic diversity of *Agave* species is crucial to secure its evolutionary processes and adapt to a changing environment. Most species of *Agave* exhibit sexual reproduction and are usually cross-pollinated, contributing to greater genetic diversity in the wild. However, many species also exhibit asexual reproduction, which can also be useful if there are few pollinators available, but it could be detrimental to genetic diversity in agave crops [23,30,31].

The objective of this study is to describe an updated compilation of the species of both subgenera of the genus *Agave* (*Agave and Littaea*) found in Mexico, analyzing available data for each species on its uses, biology and extinction risk, and suggesting alternative strategies to avoid extinction or detrimental uses of particular species.

## 2. Methodology and Analysis

A review using different digital searches, with emphasis in databases, dissertations and literature in Spanish, several English digital researches, Howard Scott Gentry’s physical *Agaves of Continental North America* book and previous reviews published on the *Agave* species found in Mexico divided by the two subgenera, *Agave* and *Littaea*, was carried out using the following criteria: distribution in Mexico, common names, pollinators and floral visitors, common uses, type(s) of reproduction, endemism, extinction risk according to the IUCN Red List [34], subspecies, and a calculation was made, based on the total importance of each species supported by the parameters mentioned above. The five criteria regarding the importance score of each species have similar weight and are explained below. It is important to emphasize that although the species from the subgenus *Littaea* are of less economic importance than the subgenus *Agave*, it is relevant to include them in the analysis because of their biological importance.

### 2.1. Risk Level Based on the IUCN Red List

CR: Critically Endangered;EN: Endangered;VU: Vulnerable;EW: Extinct in the Wild;

### 2.2. Endemism in Mexico

E: Endemic;NE: Not endemic;

### 2.3. General Uses

The species has anthropogenic use;The species has no uses;

### 2.4. Knowledge of Pollinator, Floral Visitor and/or Reproduction

Assuming that less knowledge puts the species at greater risk than more known species

Index: IUCN Red List Abbreviations for Table 1 and Table 2.
**IUCN Red List**NT: Near ThreatenedVU: VulnerableEN: EndangeredCR: Critically EndangeredEW: Extinct in the Wild

Statistical analyzes and figures were made with R y4.2.1 [35]. Graphics were made with the package ggplot 2 [36]. A summary of the analysis of both subgenera is listed below with Table 1 showing the subgenus *Agave* and Table 2 the subgenus *Littaea*. The last column titled Importance Score is based on five criteria obtained from the analysis, which are considered to be very relevant regarding the conservation status and biological environment of each species; the higher the number, the greater the extinction risk a species faces.

**Table 1 plants-12-00124-t001:** Species of the subgenus Agave with a summary of specific data according to their conservation status and biological environment.

Species	Common Names	Uses	Pollinator	Floral Visitor	Reproduction	Distribution	IUCN Red List Category	Importance Score	References
***Agave abisaii ** A. Vázquez and Nieves**		Medical use (anti-inflammatory)	*Leptonycteris yerbabuenae Choeronycteris mexicana*		Sexual: Pollination	** *Jalisco* **	EN	7	[34,37,38]
***Agave aktites* Gentry**		Food				Sinaloa, Sonora	VU	4	[4,34,38]
***Agave americana* L.**	Americano, Arroqueño (Oaxaca), Blanco, Castilla (Oaxaca), Cenizo (Tamaulipas), De Pulque (Oaxaca), Ruqueño (Oaxaca), Serrano, Sierra Negra (Oaxaca), T’ax’uada (otomi), Teometl (náhuatl), Yavi-Cuan (mixteco), Agave amarillo, Maguey serrano, Maguey cebra, Maguey cenizo, Maguey chichimeco, Maguey chino, Maguey pinto	Pulque production, distilled beverage production, ornament, textile source, food	*Leptonycteris yerbabuenae, Leptonycteris nivalis Choeronycteris mexicana*		Asexual: clonal sexual: pollination	Chihuahua, Oaxaca, Coahuila, Jalisco, South EU (Texas)		3	[4,6,13,38,39]
***Agave andreae* Sahagún and A. Vázquez**	Maguey de Piedra	Ornament				** *Michoacán* **	VU	7	[34,37,38,39]
***Agave angustifolia* Haw.**	Espadilla (Puebla), Espadín (Oaxaca), Ixtero verde, Amole, Bacanora, Chacaleño (Durango), Chelem (maya), Cincoañero (Oaxaca), Delgado (Guerrero, Oaxaca), Doba-yej (zapoteco), Gubuk (Chihuahua y Durango), Gusime (Chihuahua), Guvúkai (Chihuahua y Durango), Hamoc (seri), Juya-cuul, Ki’mai (Chihuahua y Durango), Kuúri (Chihuahua), Lineño (Jalisco), Maguey de Campo, Pelón Verde (Oaxaca), Tepemete (Durango), Yavi-incoyo, Zapupe, Henequén, Maguey de flor, Maguey de ixtle	Distilled beverage production, fiber source	*Leptonycteris yerbabuenae Choeronycteris mexicana*	*Mimus saturninus*, woodpeckers, hummingbirds, night moths	Sexual: pollination asexual: clonal bulbils	Sonora, Sinaloa, Nayarit, Jalisco, Guerrero, Oaxaca, Tamaulipas, Veracruz, Costa Rica		4	[4,38,39,40,41,42,43,44]
***Agave antillarum* Descourt.**	Maguey Antillano					Dominican Republic		2	[38,45]
***Agave applanata* Lem. ex Jacobi**	Ki’may, Maguey blanco, Maguey de Castilla, Maguey de ixtle	Pulque production, medicinal use, fiber source, food			Sexual: seeds	Chihuahua, Durango, Oaxaca, Puebla, Tlaxcala, Veracruz		3	[4,13,38,40]
***Agave asperrima* Jacobi**	Maguey Bruto	Food, mead production, distilled beverage production, fiber source			Asexual: clonal sexual: seeds	Coahuila, Durango, Nuevo León, Querétaro, San Luis Potosí, Tamaulipas		3	[38,40,41,42,45]
***Agave atrovirens* Karw.**	Agave pulquero, Flor de jiote, Flor de maguey, Flor de mezcal, Flor de pitol, Flor de quiote, Flor de sotol, Maguey de montaña, Maguey de pulque, Maguey manso, Maguey de cumbre, Tepeme	Mezcal production, fiber source, living fence			Asexual: clonal sexual: seeds	Puebla, Oaxaca		3	[4,13,38]
***Agave aurea* Brandegee**	Lechuguilla, Lechuguilla mezcal, Maguey	Ornament, fiber source			Asexual: clonal	** *Baja California Sur* **		5	[38,44,46,47]
***Agave avellanidens* Trel.**		Ornament			Asexual: clonal	** *Baja California* **	NT	5	[34,38,46,47]
***Agave azurea* R. H. Webb and G. D. Starr**					Sexual: seeds	** *Baja California Sur* **	VU	4	[34,38,47]
***Agave bovicornuta* Gentry**	Lechuguilla de la Sierra (Sonora), Masparillo (Durango) Cerial	Ornament, food, mezcal production (not so common), fiber source			Sexual: pollination, seeds	Sonora, Sinaloa, Chihuahua	VU	4	[4,34,38,39,40]
** *Agave cantala* ** **(Haw.) Roxb. ex Salm-Dyck**	Cincoañero (Oaxaca), Maguey del Cinco (Oaxaca) Henequén	Fiber source, fistilled beverage production			Asexual: clonal	Oaxaca, Jalisco		3	[4,34,38,39,40]
***Agave capensis* Gentry**	Mescalito	Food				** *Baja California* **	EN	8	[4,34,38,39,40]
***Agave chrysantha* Peebles**		Food, fiber source	*Leptonycteris yerbabuenae, Apis mellifera*	Bats, birds, insects	Sexual: pollination seeds asexual: clonal	Sonora, Southwest EU (Arizona)		2	[30,38,40,42,44]
***Agave colorata* Gentry**	Ceniza (Sonora), Haamjö, Caacöl	Food, mead production, mezcal production, ornament	*Leptonycteris*, hummingbirds	Hummingbirds	Sexual: pollination, seeds asexual: clonal	Sinaloa, Sonora		2	[4,38,39,40,44]
***Agave congesta* Gentry**	Maguey Tzotzil	Ornament			Sexual: Seeds	** *Chiapas* **		6	[4,38,40,48,49,50]
***Agave cupreata* Trel. and A. Berger**	Papalometl, Papalote (Guerrero), Ancho, Chino (Michoacán), Cimarrón, Tuchi, Yaabendisi (mixteco), Maguey bravo, Tobalá	Mezcal production, food, mead production			Sexual: seeds	Guerrero, Michoacán	EN	5	[34,38,39,40,41]
***Agave datylio* F. A. C. Weber**		Ornament			Asexual: clonal	** *Baja California Sur* **		5	[4,38,44,47]
***Agave decipiens* Baker**					Asexual: clonal	Southeast EU (Florida), Yucatán	VU	2	[4,34,38,45]
***Agave delamateri* W. C. Hodgs and Slauson**		Food, fiber source		Birds, insects	Asexual: clonal	** *Southwest EU (Arizona)* **		5	[38,51]
***Agave deserti (complex)* Engelm.**	Maguey del desierto	Ornament, food, fiber source, mead source		Hummingbirds, insects	Sexual: seeds asexual: clonal	Baja California Sonora Southwest EU		3	[4,38,40,44,47,52]
***Agave deserti Agave cerulata (complex)* Trel.**	Mescal	Fiber source, food, Mead production, distilled beverage production				Baja California, Sonora		4	[38,40,44,47]
***Agave deserti Agave subsimplex (complex)* Trel.**	A’amxw	Food	*Bombus, Leptonycteris yerbabuenae, Lepidoptera*	Bats	Sexual: pollination	** *Sonora* **	VU	5	[34,38,40,44,53]
***Agave desmettiana* Jacobi**	Maguey de pita	Ornament, mead production, distilled beverage production	*Glossophaga soricina*		Sexual: pollination asexual: clonal	Sinaloa, Southwest EU (California)		3	[4,38,40,54]
***Agave durangensis* Gentry**	Cenizo (Durango), Bayuza	Ornament, mezcal production, food			Sexual: seeds asexual: clonal	Durango, Zacatecas		3	[4,38,39,40,55]
***Agave flexispina* Trel.**		Ornament				Chihuahua, Durango, Zacatecas	VU	5	[4,34,38]
***Agave fortiflora* Gentry**	Haamjö, Caacöl	Food, mead production			Asexual: clonal	** *Sonora* **		5	[4,38,40]
***Agave fourcroydes* Lem.**	Henequén, Jenequén, Maguey sisal	Fiber source, ornament, food			Asexual: clonal nulbils	Yucatán, Cuba		3	[4,38,40,55]
***Agave gentryi* B. Ullrich**	Maguey del Bosque	Food, mead production, fiber source	Chiroptera		Sexual: pollination	Nuevo León, Coahuila, Tamaulipas, Durango, Zacatecas, San Luis Potosí, Hidalgo, Mexico, Puebla		3	[38,40,56,57]
***Agave gigantensis* Gentry**		Ornament, mezcal production			Sexual: seeds	** *Baja California Sur* **		5	[4,38,40,46]
***Agave gracilipes* Trel.**	Maguey de Pastizal					Chihuahua, South EU (Texas), Nuevo Mexico		2	[4,38]
***Agave grijalvensis* B. Ullrich**	Maguey del Grijalva	Ornament, food			Sexual: seeds	** *Chiapas* **	EN	7	[34,38,58]
***Agave guadalajarana* Trel.**	Mascarreño	Distilled beverage production, ornament, medical use			Sexual: seeds	** *Jalisco* **	EN	7	[4,34,38,40,42]
***Agave gypsophila* Gentry**	Maguey de Ixtli, quiote	Used for rural house construction			Asexual: clonal	** *Guerrero* **	CR	9	[34,37,38,49,50]
***Agave havardiana* Trel.**	Maguey norteño	Mezcal production	*Leptonycteris nivalis, Choeronycteris mexicana*	*Zenaida asiatica, Icterus parisorum*	Sexual: pollination	Chihuahua, Coahuila, South EU (Texas)	VU	3	[34,38,59,60]
***Agave hiemiflora* Gentry**		Wild			Sexual: seeds	Chiapas, Guatemala		1	[4,38,48]
***Agave hookeri* Jacobi**	Maguey Ixquitécatl	Pulque production, food, fiber source, living fence			Sexual: seeds asexual: clonal	** *Michoacán* **		5	[4,13,38,40]
***Agave inaequidens* K. Koch**	Hocimetl (náhuatl), Largo (Michoacán), Lechuguilla	Pulque production, mezcal production, food, fiber source	*Leptonycteris yerbabuenae Icterus bullockii*	*Apis mellifera, Vespidae, Bombus*, night moth, hummingbirds, hooded warbler	Asexual: clonal sexual: pollination seeds	Jalisco, Michoacán, Colima		2	[13,38,39,40,42]
***Agave isthmensis* A. García-Mend. and F. Palma**	Maguey Istmeño	Ornament				Chiapas, Oaxaca	VU	5	[34,38]
***Agave jaiboli* Gentry**		Food, mezcal production, fiber source			Sexual: seeds	Sonora, Chihuahua	VU	4	[4,34,38,40]
***Agave karwinskii* Zucc.**	Al-mal-bi-cuish (chontal), Barril: Verde/Amarillo/Blanco, Bicuixe (Oaxaca), Cachutum (popolca), Cirial (Oaxaca), Cuishi (Oaxaca), Dob-cirial, Madrecuixe (Oaxaca), Manso, San Martinero (Oaxaca), Tobasiche (zapoteco/Oaxaca), Tripón (Oaxaca), Verde (Oaxaca), Candelilla, Candelillo, Canelillo corazón, Espadilla, Cuishe, Greñudo, Cuish, Madre cuish	Mezcal production, ornament, living fence, fiber source, food	*Leptonycteris yerbabuenae*		Sexual: pollination asexual: clonal	Oaxaca, Puebla, Veracruz	VU	4	[4,34,38,39,40,61]
***Agave kewensis* Jacobi**	Maguey del Grijalva	Food			Asexual: clonal	** *Chiapas* **		6	[38,40,48,49,50]
***Agave kristenii* A. Vázquez and Cházaro**	Maguey de Piedra	Medical use, ornament			Asexual: clonal	** *Michoacán* **	CR	8	[34,37,38]
***Agave lexii ** García-Mor., García-Jim. and Iamonico**		Wild				** *Tamaulipas* **		4	[38,62]
***Agave lurida* Aiton**	Maguey de la Luna	Ornament, food				** *Oaxaca* **	EW	13	[4,34,38,49,50]
***Agave lyobaa* García-Mend. and S. Franco**	Maguey Coyote	Mezcal production				Oaxaca, Puebla	EN	6	[63]
***Agave macroacantha* Zucc.**	Barril Verde (Oaxaca), Cincoañero (Oaxaca), Cachrolochje’, Espadilla, Estafalalate	Ornament, mead production, distilled beverage production, fiber source	*Leptonycteris yerbabuenae Choeronycteris mexicana Lepidoptera (Noctuidae, Sphingidae, Microlepidoptera*	Bats, *Colaptes auratus*, hummingbirds	Sexual: pollination asexual: clonal bulbils	Oaxaca, Puebla	EN	4	[34,38,39,40,42,61,64,65]
***Agave macroculmis* Tod.**	Maguey verde	Food, mead production			Asexual: clonal	Chihuahua, Coahuila		3	[4,38,66]
***Agave mapisaga* Trel.**	Aguamiel, Maguey manso, Maguey pulquero	Pulque production, distilled beverage production, food, fiber source, ornament, mead production			Asexual: clonal sexual: pollination	Chihuahua, Tlaxcala, Puebla, Hidalgo		3	[13,38,40,41,66]
***Agave margaritae* Brandegee**		Distilled beverage production			Asexual: clonal	** *Baja California Sur* **		5	[38,40,47]
***Agave marmorata* Roezi**	Tepeztate o Tepextate (Oaxaca), Curandero, Lechuguilla, Maguey de Caballo, Pitzometl (náhuatl/Puebla), Tdu-cual ó Du-cual (zapoteco), Tecolote, Pichomel, Pitzomel, Pichometl	Mezcal production, food, mead production, fiber source	*Leptonycteris yerbabuenae Choeronycteris mexicana*	Bats, *Colaptes auratus*, hummingbirds	Sexual: pollination, seeds	Oaxaca, Puebla, Tlaxcala		2	[4,38,39,40,53,61]
***Agave maximiliana* Baker**	Lechuguilla (Jalisco), Manso, Masparillo (Durango), Tecolote, Raicilla	Mezcal production, mead production, food, medical use				Jalisco, Nayarit, Durango, Southwest EU (California, Arizona)		4	[4,38,39,40,67]
***Agave mckelveyana* Gentry**			Hummingbirds, *Xylocopa*, wasps		Sexual: pollination asexual: clonal	** *Southwest EU (Arizona)* **		3	[4,38,44]
***Agave montana* Villareal**		Ornament, mead production, fiber source				Tamaulipas, Nuevo León, Coahuila		4	[38,40]
***Agave moranii* Gentry**		Food, fiber source				** *Baja California* **		6	[4,38,46]
***Agave murpheyi* Gibson**	Agave hohokam	Food, distilled beverage production, fiber source			Sexual: seeds asexual: clonal bulbils	Sonora, Southwest EU (Arizona)		3	[4,38,40,51]
***Agave nayaritensis* Gentry**						Sinaloa, Durango, Nayarit	VU	3	[4,34,38,48]
***Agave oroensis* Gentry**						Zacatecas		2	[4,38,40,48]
***Agave ovatifolia* G. D. Starr and Villareal**		Ornament, fiber source				Nuevo León		4	[38,40,48]
** *Agave pablocarrilloi A. Vázquez, Muñiz-Castro and Padilla-Lepe* **						** *Colima* **		4	[37,38]
***Agave pachycentra* Trel.**						Oaxaca, Chiapas, Guatemala		2	[4,38,48]
***Agave palmeri* Engelm.**	Lechuguilla (Sonora)	Food, fiber source, distilled beverage production	*Bombus*, *Leptonycteris nivalis, Choeronycteris mexicana*	Hummingbirds, insects	Sexual: pollination	Southwest EU (Arizona), Sonora		2	[4,30,38,39,40]
***Agave parrasana* A. Berger**	Noa	Ornament, fiber source				** *Coahuila* **	VU	8	[4,34,38,40,49,50]
***Agave parryi* Engelm.**	Maguey mezcal	Mezcal production, food, fiber source, ornament			Asexual: clonal	Chihuahua, Durango, Southwest EU (Nuevo Mexico and Arizona)		3	[4,38,40]
***Agave phillipsiana* W. C. Hodgs.**		Food			Asexual: clonal	Southwest EU (Arizona)		3	[38,68]
***Agave potatorum* Zucc.**	Tobalá o Dob-ala (Zapoteco/Oaxaca), Biliá (Oaxaca), Dob-be, Maguey de Monte, Papalometl (Puebla/Oaxaca), Yauiticuxi (mixteco), Arruqueño, Magueycillo	Mezcal production, food, ornament	*Leptonycteris yerbabuenae*, *Leptonycteris nivalis*, *Apis mellifera*, *Bombus* sp., *Tabanus* sp.	Bats, birds	Asexual: clonal (occasional) Sexual: pollination, seeds	Oaxaca, Puebla	VU	3	[4,34,38,39,40,42,69]
***Agave pringlei* Engelm. ex Baker**					Asexual: clonal	** *Baja California* **		3	[38,47]
***Agave promontorii* Trel.**		Ornament			Sexual: seeds	** *Baja California Sur* **	CR	8	[4,34,38,47]
***Agave rhodacantha* Trel.**	Cimarrón amarillo, Maguey de campo, Ixtéro amarillo	Mezcal production, fiber source, ornament, living fence			Sexual: seeds	Sonora, Nayarit, Jalisco, Oaxaca		3	[38,40,70]
***Agave salmiana* Otto ex Salm-Dyck**	Amarillo (Puebla), Bronco Mbänuada (otomí), Cimarrón, Del Valle (Oaxaca), Doba gashon ó Doba lash (Oaxaca), Llano (Oaxaca), Maguey de Pulque, Manso, Potosino, Verde (San Luis Potosí), Xagarcia (Oaxaca) Maguey pamilla, Maguey pinto	Pulque production, mead production, mezcal production, food (“Pan de Pulque”) fiber source, ornament	*Leptonycteris yerbabuenae, Leptonycteris nivalis, Choeronycteris mexicana, Bees, Hummingbirds*	Bats, birds	Asexual: clonal sexual: pollination seeds	Puebla, Hidalgo, Chihuahua, Jalisco, Tlaxcala, Southwest EU (California and Arizona)		2	[4,13,15,38,39,41,42,56]
***Agave scabra* Ortega**	Lamparillo	Ornament	*Leptonycteris yerbabuenae, Leptonycteris nivalis*		Sexual: pollination asexual: clonal	Chihuahua, Coahuila		3	[4,38,56,59]
***Agave scaposa* Gentry**	Maguey de Macho	Used for construction, food, living fence	*Leptonycteris* sp.	*Bombus*, hummingbirds	Sexual: pollination asexual: clonal	Puebla, Oaxaca		2	[4,38]
***Agave sebastiana* Greene**		ornament			Asexual: clonal	** *Baja California Sur* **		5	[4,38,47]
***Agave seemanniana* Jacobi**	Biliaa, Maguey chato	Mezcal production, food			Sexual: seeds	Oaxaca, Chiapas, Nicaragua		5	[4,38,40]
***Agave shawii* Engelm.**	Amal	Ornament, food, distilled beverage production		Hummingbirds *Leptonycteris*, *Bombus*	Sexual: seeds asexual: clonal	Baja California, Southwest EU		3	[38,40,44,47]
***Agave shrevei* Gentry**	Lechuguilla (Sonora), Ceniza, Lechuguilla ceniza, Mezcal blanco	Mezcal production, mead production, food				** *Sonora* **		6	[4,38,39,40,71]
***Agave sisalana* Perrine**	Henequén, Henequén verde, Maguey tuxtleco, Sisal	Fiber source, ornament			Asexual: clonal bulbils	Yucatán, Chiapas		3	[4,38,40,55]
***Agave sobria* Brandegee**	Lechuguilla, Mezcal pardo	Mezcal production, food, ornament			Sexual: seeds asexual: clonal	** *Baja California Sur* **		5	[38,40,44,46,47]
***Agave stringens* Trel.**						** *Jalisco* **		4	[4,38,45]
***Agave tequilana* F. A. C. Weber**	Azul (San Luis Potosí, Jalisco), Chato (Michoacán), Tequila, Agave tequilero, Bermejo, Mano larga, Pata de mula, Chino azul, Chino bermejo	Tequila production, Food, fiber source, ornament, soap production			Asexual: clonal Sexual: pollination (minimum)	Jalisco		3	[4,38,39,40,41,55]
***Agave turneri* R. H. Webb and Salazar-Ceseña**					Asexual: clonal	** *Baja California* **	EN	5	[34,38,47]
** *Agave valenciana* ** **Cházaro and A. Vázquez**	Relisero (Jalisco)	Mezcal production, raicilla production, ornament			Sexual: seeds	** *Jalisco* **	CR	8	[34,39,40,72]
***Agave vivipara* L.**	Lechuguilla	Food, mead production, fiber source, distilled beverage production					VU	5	[34,38,40]
***Agave vizcainoensis* Gentry**	Maguey de El Vizcaíno	ornament	*Leptonycteris yerbabuenae*		Asexual: clonal Sexual: pollination	** *Baja California Sur* **		6	[38,47,49,50]
***Agave weberi* J. F. Cels ex J. Poiss.**	Maguey mezcalero, Maguey verde	Mezcal production, pulque production, ornament, fiber source			Asexual: clonal Sexual: pollination, seeds	Chihuahua, Coahuila, Durango, San Luis Potosí, Tamaulipas		3	[4,13,38,40]
***Agave wocomahi* Gentry**	Maguey verde, Ojcome	Mezcal production, food, fiber source, mead production	*Leptonycteris yerbabuenae*		Sexual: pollination, seeds	Sonora, Durango, Chihuahua, Jalisco		3	[4,38,40]
***Agave x glomeruliflora* (Engelm.) A. Berger**						Durango, Coahuila		2	[38,73]
***Agave zebra* Gentry**	Áamxw, Káokt’	Ornament, mezcal production, food			Sexual: seeds Asexual: bulbils	** *Sonora* **	VU	6	[34,38,40,44,74]

* Distribution both in *Italics* and **Bold** means the species is endemic from that state.

**Table 2 plants-12-00124-t002:** Species of the subgenus Littaea with a summary of specific data according to their conservation status and biological environment.

Species	Common Names	Uses	Pollinator	Floral Visitor	Reproduction	Distribution	IUCN Red List Category	Importance Score	References
***Agave albomarginata* Gentry**	Maguey de márgenes claros	Ornament			Asexual: clonal	Coahuila, Chihuahua, Nuevo León, Querétaro, San Luis Potosí, Tamaulipas	EN	5	[4,34,38,40,48]
***Agave albopilosa* I. Cabral, Villareal and A. E. Estrada**	Maguey viejito	Fiber source, ornament				** *Sierra Madre Oriental (exact location not given for protection)* **	CR	9	[34,38,40,75]
***Agave angustiarum* Trel.**	Lechuguilla suave, Maguey de ixtle, Cacaya	Fiber source, food				Guerrero, Michoacán, Morelos, Oaxaca, Puebla		4	[38,48,76]
***Agave arizonica* Gentry and J. H. Weber**		Ornament			Asexual: clonal	** *Southwest EU (Arizona)* **		5	[4,38]
** *Agave arcedianoensis* ** **Cházaro, O. M. Valencia and A. Vázquez**	Maguey de Arcediano	Ornament				** *Jalisco* **	VU	7	[34,38,48,61]
***Agave attenuata* Salm-Dyck**	Maguey del Dragón	Ornament			Asexual: clonal	Guerrero, Jalisco, Colima, Michoacán, Nayarit, Sinaloa, Durango		3	[4,38,77]
***Agave bakeri* H. Ross**		Ornament			Asexual: clonal Sexual: seeds			3	[4,38]
***Agave bracteosa* S. Watson ex Engelm.**	Maguey araña	Ornament, fiber source			Asexual: clonal	** *Coahuila* **		7	[4,38,40,49,50]
***Agave calciphila* G. D. Starr**						** *Oaxaca* **		4	[38,78]
***Agave celsii* Hook.**	Maguey comezonudo, Maguey de Peña	Medical use	*Bombus, Leptonycteris*, hummingbirds *Sphingidae*			Hidalgo, Puebla, Querétaro, San Luis Potosí, Tamaulipas		3	[38,79]
** *Agave chazaroi* ** **A. Vázquez and O. M. Valencia**		Ornament				** *Jalisco* **	VU	7	[34,38,76,77]
***Agave chiapensis* Jacobi**	Maguey Chamula	Ornament, food				Chiapas, Oaxaca	VU	8	[4,34,38,40,49,50]
***Agave chrysoglossa* I. M. Johnst.**	Hasot, Amole	Ornament, local use for clothes washing, food	*Leptonycteris yerbabuenae*, hummingbirds	Insects, hummingbirds	Sexual: pollination, seeds Asexual: clonal	** *Sonora* **		4	[4,38,40,44]
***Agave colimana* Gentry**	Maguey de Colima	Ornament				Colima, Jalisco, Nayarit		4	[4,38,48]
***Agave convallis* Trel.**	Jabalí (Oaxaca), Maguey Escobeta	Distilled beverage production				** *Oaxaca* **	VU	7	[34,38,39,48,67,80]
***Agave dasyliriodes* Jacobi and C. D. Bouche**	Maguey intrépido	Ornament			Sexual: Seeds	** *Morelos* **	EN	9	[4,34,38,48,49,50]
***Agave difformis* A. Berger**	Lechuguilla	Ornament, fiber source, soap	*Leptonycteris yerbabuenae, Leptonycteris nivalis, Choeronycteris mexicana*	*Apis mellifera, Lasioglossum lasioglassum, Bombus, Centris, Polistinae, Agrius cingulatus, Pachylia ficus, Sphinx lugens, Erinnyis ello*	Sexual: pollination Asexual: clonal	San Luis Potosi, Hidalgo		2	[4,38,81]
***Agave doctorensis* L. Hern. and Magallán**						** *Querétaro* **	CR	7	[34,38,82]
***Agave ellemeetiana* Jacobi**					Asexual: clonal Sexual: seeds	Veracruz, Oaxaca		1	[4,38,83]
***Agave felgeri* Gentry**		Food			Asexual: clonal	** *Sonora* **	VU	8	[4,38,40,44]
***Agave filifera* Salm-Dyck**	Amole, Maguey de maceta	Pulque production, fiber source			Asexual: clonal Sexual: pollination	Hidalgo, Morelos		3	[4,13,38,40]
***Agave garcia-mendozae* Galván and Hern.**		Ornament, fiber source	*Leptonycteris yerbabuenae, Leptonycteris nivalis, Choeronycteris mexicana*	*Apis mellifera, Agrius cingulatus, Pachylia ficus, Sphinx lugens, Erinnyis ello*	Sexual: pollination	** *Hidalgo* **	VU	5	[34,38,81]
***Agave garciaruizii* A. Vázquez, Hern.-Vera and Padilla-Lepe**					Asexual: clonal	Jalisco, Michoacán		1	[38,76]
***Agave geminiflora* (Tagl.) Ker Gawl.**	Palmilla	Ornament, fiber source	Hummingbirds		Sexual: pollination	Jalisco, Nayarit	VU	4	[4,34,38,40]
***Agave ghiesbreghtii* Lem. ex Jacobi**		Living fence			Asexual: clonal	Oaxaca, Puebla		3	[4,38]
***Agave glomeruliflora* (Engelm.) A. Berger**		Ornament			Sexual: seeds	Coahuila		3	[4,38]
***Agave gracielae* Galvan and Zamudio**		Food, ornament				Querétaro, San Luis Potosí		4	[38,84]
***Agave guiengola* Gentry**	Maguey plateado	Ornament			Asexual: clonal Sexual: seeds	** *Oaxaca* **	EN	9	[4,34,38,48,49,50]
***Agave gypsicola* García-Mend. and D. Sandoval**	Maguey blanco (xavi kuiji)	Food, living fence				** *Oaxaca* **		6	[45,63]
** *Agave horrida* ** **Lem. ex Jacobi**	Maguey bueno	Ornament, food, fiber source, distilled beverage production	*Leptonycteris nivalis Choeronycteris mexicana*	*Tegeticula, Apis mellifera*	Sexual: pollination Seeds	** *Morelos* **		4	[4,38,40,59,85]
***Agave impressa* Gentry**	Maguey Masparrillo	Medical use			Asexual: clonal	** *Sinaloa, Nayarit* **	EN	9	[4,34,38,49,50,61]
***Agave jimenoi* Cházaro and A. Vázquez**						** *Veracruz* **		4	[38,86]
***Agave kavandivi* García-Mend. and C. Chávez**		Food				** *Oaxaca* **	CR	9	[34,38,87]
***Agave kerchovei* Lem.**	Cacalla, Rabo de León	Food, fiber source, mead production, distilled beverage production	*Leptonycteris yerbabuenae*		Sexual: pollination	Puebla, Oaxaca	VU	4	[4,34,38,40,61]
***Agave lechuguilla* Torr.**	Amole, Lechuguilla, Istle, Ixtle	Fiber source	*Hyles lineata*, *Xylocopa californica*, *Bombus pennsylvanicus*, *Eugenes fulgens*, *Calothorax lucifer*, *Archilochus alexandri*, *Selasphorus* sp.	*Apis mellifera*, *Selasphorus* sp., vespidae, small bees	Asexual: clonal Sexual: pollination	Hidalgo, San Luis Potosí, Tamaulipas, Coahuila, Chihuahua, Texas, Nuevo Mexico		2	[4,30,38,40,88]
***Agave lophantha* Schiede**	Maguey Estoquillo	Ornament				South EU (Texas), Chihuahua, Veracruz		4	[4,38]
***Agave manantlanicola* Cuevas and Santana-Michel**					Asexual: clonal	** *Jalisco* **	EN	5	[34,38,77]
***Agave maria-patriciae ** Cházaro and Arzaba**						** *Veracruz* **		4	[38,89]
***Agave megalodonta ** García-Mend. and D. Sandoval**	Maguey espumoso	Occasional mezcal production				Oaxaca, Puebla, Guerrero	NT	4	[63]
***Agave mitis* Mart.**	Maguey de Peña	Food, fiber source		*Chiroptera, Bombus, Sphingidae,* hummingbirds		Guanajuato, Coahuila, Puebla		3	[38,40,90]
***Agave montium-sancticaroli* García-Mend.**	Jarcia	Mezcal production, food			Asexual: clonal	** *Tamaulipas* **	CR	8	[34,38,40,73]
***Agave multifilifera* Gentry**	Chahuí	Ornament, fiber source, food, mead production, distilled beverage production				Chihuahua, Sinaloa, Nayarit, Sinaloa		4	[4,38,40]
***Agave muxii ** Zamudio and G. Aguilar-Gutiérrez**					Asexual: clonal	Querétaro, San Luis Potosí		1	[91]
***Agave nizandensis* Cutak**	Maguey de Nizanda	Ornament				** *Oaxaca* **	CR	12	[4,34,38,49,50]
***Agave nussaviorum* García-Mend.**	Maguey Papalometl	Food, medical use, construction material, forage, distilled beverage production				** *Oaxaca* **		6	[38,40,69]
***Agave obscura* Schiede ex Schltdl.**	Lechuguilla Bronca	Ornament			Asexual: clonal	Oaxaca, Puebla, San Luis Potosí, Tamaulipas, Veracruz		3	[4,55]
***Agave ocahui* Gentry**	Amolillo	Ornament, fiber source			Sexual: pollination seeds	** *Sonora* **	NT	5	[4,34,38,40]
***Agave oteroi ** G. D. Starr and T. J. Davis**						Oaxaca, Puebla		2	[38,92]
***Agave ornithobroma* Gentry**	Maguey Pajarito	Fiber source		*Psitaciformes*	Asexual: clonal	Sinaloa, Nayarit	VU	5	[4,34,38,49,50]
***Agave parviflora* Torr.**	Tauta (Sonora)	Ornament, food	*Bombus sonorus, Xylocopa*		Sexual: pollination Asexual: clonal	Chihuahua, Sonora, Southwest EU, Argentina		5	[4,30,38,39,40,49,50,81,93]
***Agave peacockii* Croucher**	Amol, Lechuguilla, Maguey de Ixtle, Maguey fibroso	Fiber source, food, mead production, distilled beverage production			Asexual: clonal	Hidalgo, Puebla, Oaxaca	VU	5	[4,34,38,40,49,50]
***Agave pedunculifera* Trel. ex Standll.**	Lechuguilla	Ornament			Asexual: clonal	Sinaloa, Jalisco, Nayarit, Colima, Michoacán, Guerrero		3	[4,38]
***Agave pelona* Gentry**	Bacanora, Mezcal pelón	Food, distilled beverage production, ornament, fiber source	*Leptonycteris yerbabuenae*		Sexual: pollination	** *Sonora* **	CR	8	[4,34,38,40,44,78]
***Agave pendula* Schnittsp.**		Ornament			Asexual: clonal	Veracruz, Chiapas, Oaxaca		3	[4,38,48]
***Agave petrophila* A. García-Mend. and E. Martínez**		Ornament				Oaxaca, Guerrero	EN	6	[34,38,84]
***Agave polianthiflora* Gentry**	Chahuí	Ornament, food, mead production			Asexual: clonal	Sonora, Chihuahua		5	[4,38,40,49,50]
***Agave polyacantha* Haw.**						Tamaulipas, Veracruz		2	[4,38]
***Agave potrerana* Trel.**		Ornament				Coahuila, Chihuahua		4	[4,38]
***Agave quiotepecensis* García-Mend. and S. Franco**	Agave Rabo de León	Fiber source, food, forage				** *Oaxaca* **	NT	6	[63,78]
***Agave rzedowskiana* P. Carrillo, Vega and R. Delgad.**						Sinaloa, Jalisco		2	[38,94]
***Agave schidigera* Lem.**	O’r, Lechuguilla Mansa	Fiber source				Chihuahua, Jalisco, Zacatecas, Sinaloa, Durango, Aguascalientes, Guerrero, San Luis Potosí, Michoacán		4	[4,38,40]
***Agave schottii* Engelm.**	Icapánim, Amole	Leaves used as clothing soap, food, mead production	*Bombus, Xylocopa, Leptonycteris*	Hummingbirds, *Xylocopa, Apis*	Sexual: pollination Asexual: clonal	Northwest Chihuahua, Sonora, Southwest and South EU (Arizona and Nuevo Mexico)		2	[4,38,40,44,81,93]
***Agave striata* Zucc.**	Junquillo, Estoquillo, Maguey espadín, Palmita, Peinecillo	Ornament, fiber source, mead production	*Leptonycteris yerbabuenae, Leptonycteris nivalis, Choeronycteris mexicana*	*Apis mellifera, Lasioglossum lasioglassum, Bombus, Centris, Polistinae, Eugenes fulgens, Cynantus latirostris, Agrius cingulatus, Pachylia ficus, Sphinx lugens, Erinnyis ello*	Sexual: pollination Asexual: clonal	Coahuila, Nuevo León, Tamaulipas, Durango, Zacatecas, San Luis Potosí, Queretaro, Puebla		2	[4,30,38,40,81]
***Agave stricta* Salm-Dyck**	Pelo de ángel	Ornament, fiber source, food				Puebla, Oaxaca		4	[4,38,40]
***Agave tenuifolia* Zamudio and E. Sánchez**	Maguey de la Sierra Madre Oriental					Querétaro, Tamaulipas, Hidalgo, San Luis Potosí		2	[38,95]
***Agave titanota* Gentry**	Cachitún	Ornament, distilled beverage production			Sexual: seeds	** *Oaxaca* **	EN	8	[4,34,38,40,49,50]
***Agave triangularis* Jacobi**	Cacalla, Maguey tunecho	Ornament, food, fiber source			Asexual: clonal	Puebla, Oaxaca	VU	4	[4,34,38,40]
***Agave univittata* Haw.**	Estoquillo, Lechuguilla (Sonora), Mezortillo	Distilled beverage production, fiber source						4	[4,38,39,40]
***Agave vazquezgarciae* Cházaro and J. A. Lomelí**		Food				** *Jalisco* **		6	[38,77,96]
***Agave victoriae-reginae (complex)* T. Moore**	Noa	Food, Fiber source, distilled beverage production			Asexual: clonal	Chihuahua, Coahuila, Nuevo León, Durango		6	[4,38,40,49,50,55,97]
** *Agave victoriae-reginae (complex)* **	Maguey del Rey Fernando				Asexual: clonal	** *Coahuila* **	CR	6	[34,38,97]
***Agave nickelsiae* Rol.-Gross**									
** *Agave victoriae-reginae (complex)* **		Ornament			Sexual: seeds Asexual: rhizomes	** *Durango* **	CR	8	[34,38,97]
** *Agave pintilla* ** **S. González, M. González and L. Reséndiz**									
***Agave vilmoriniana* A. Berger**	Ahué, Amole	Ornament, mead production, food, fiber source		*Apis, Leptonycteris*, Hummingbirds	Asexual: clonal bulbils Sexual: seeds	Sonora, Sinaloa, Durango, Jalisco, Aguascalientes, Southwest EU		3	[4,38,40,44]
***Agave warelliana* De Smet ex T. Moore and Mast.**						Guatemala, Chiapas	EN	4	[4,34,38,45]
***Agave wendtii* Cházaro**						** *Veracruz* **	EN	6	[34,38,48]
***Agave xylonacantha* Salm-Dyck**	Kuat’ ma’ ye	Food, fiber source		*Bombus*, *Leptonycteris*, hummingbirds, *Sphingidae*, *Choeronycteris mexicana*	Sexual: pollination, seeds	Hidalgo, Guanajuato, Querétaro, Tamaulipas, San Luis Potosí, Nuevo León		3	[4,38,40,59]
***Agave yuccifolia* DC**		Ornament				Hidalgo		4	[4,38]

* Distribution both in *Italics* and **Bold** means the species is endemic from that state.

Table 3 is the criteria and corresponding weight, or score, of each species used to create Table 4 where the species with the highest total importance scores were placed under three possible categories mentioned below. The higher the score of the parameters in Table 3, the higher the risk of extinction for the species, therefore those species require more attention, although precautionary measures should also be taken with the rest of the species.

The species with the greatest importance score were arranged by subgenera and separated into three categories as follows: 1. Severely endangered: those species facing the possibility of extinction and that should be subjected to an urgent rescue and recovery program, considering the possibility of stopping harvest from the wild altogether. 2. High risk of extinction: these species need a recovery program and their harvest should be subjected to a strong conservation program. 3. Threatened species: management and recovery plans must be prepared and implemented for their harvest.

In order to evaluate the influence of score criteria in the Agave species, a Principal Component Analysis (PCA) was performed for each subgenus with the package FactoMineR [98], with the variables and categories used for the Importance score reported in Table 3.

## 3. Results and Discussion

The species distribution for the subgenera *Littaea* and *Agave* by state of Mexico are shown in Figure 1 and Figure 2, respectively. It is noteworthy that there are no records in Baja California or Yucatán for *Littaea*, and for *Agave* there are only three reported species in Yucatán, while for both subgenera, there are no published records for Quintana Roo, Campeche and Tabasco, which can be due to a lack of sampling in these areas. In *Littaea* the highest richness occurs in the southern state of Oaxaca, while for *Agave* also Oaxaca is the richest state, along with Sonora and Chihuahua in the northwestern part of the country. Both subgenera have lower numbers of species in the center of the country.

In both subgenera, the proportion of plants used for food is similar (*Littaea* 18.6% and *Agave* 20.7%, Figure 3), while *Agave* species are much more used for making distilled beverages (mainly mezcal and tequila) (*Littaea* 8.9% and *Agave* 18.8 %). The better-known species used for the production of distilled beverages were not considered at high risk, due to the proportionally greater knowledge available, although many of them might be endangered because of the lack of proper production management. For instance, these popular Agaves are now cultivated in massive monocultures, while preventing any sexual reproduction, and thus their genetic diversity is rapidly declining, such is the case of *A. tequilana* [6,18].

The use for fibers is proportionally similar for *Littaea* and *Agave* (18.6% vs. 16.5%, respectively, Figure 3), compared to the use of ornamentals, where *Littaea* has a higher use (28.3% vs. 18.8%, respectively, Figure 3). Species used for fiber, such as *A. parrasana*, *A. albopilosa*, *A. pelona* and *A. pintilla* were categorized at a high risk of extinction, while *A. lurida* and *A. nizandensis* are considered severely endangered and used as ornamentals. It is outstanding that in both subgenera the vast majority of the reported species have some kind of use (*Litteae* 88.9% and *Agave* 93.03%, Figure 3).

It is important to note that most of the species included in Table 4, which are in need of specific conservation efforts, are used for fiber, ornamentals or have no known uses. There is very little information regarding pollinators and visitors in both subgenera for most species (more than 75%, Figure 4). Obviously, this is important information necessary to develop adequate conservation and management strategies. Bat pollination records are much greater in *Agave* than in *Littaea* (Figure 5). For visitors, available information shows an important difference between subgenera (Figure 5), but more studies are needed to disentangle if these visitors are acting as true pollinators. There is ample information for the general type of reproduction in both subgenera (Figure 5), which may be due to the fact that as mentioned above, species from both subgenera are widely used, so there is an interest in having this information.

The PCA analyses of the different variables reported in Table 3 for both subgenera (Figure 6 and Figure 7) were divided into four quadrants:

I. Species with higher importance scores are in Quadrant I. Many of these species are endemic and have many uses, so they can be considered species of main concern for conservation and management.

II. This quadrant contains endemic species with little use and with little or no reproductive information, which is mainly associated with a lack of interest, in contrast to species in quadrants I and III.

III. Species with many uses, so much more reproductive information is available, but a lower importance score than species in Quadrant I.

IV. Species that apparently have no use and are not endemic, but reproductive information is lacking for them.

Having an understanding of the status of each species of the genus *Agave* in Mexico is fundamental to species conservation and sustainable use, and to create awareness of the practices used nowadays on the utilization of the different species of the genus, averting the possible tragic outcome of extinction. Artisanal and industrial producers are invited to maintain or opt for biodiversity-friendly alternatives for the conservation of agaves, practicing sustainable methods in their practices with agaves. Suggested approaches include the use of species considered at low risk of extinction for the production of distilled beverages (for instance *A. americana*, *A. duranguensis*, *A. karwinskii* or *A. macroacantha*) or ecofriendly production methods for agaves and their pollinators, such as the Bat Friendly Tequila and Mezcal project [99] created by UNAM (Universidad Nacional Autónoma de Mexico) and the Tequila Interchange Project [19,100]. This project seeks to preserve genetic diversity, seed production and pollinator populations, including bats, by allowing 5% of the agaves in agave fields destined for production of distillates to flower and produce seeds.

Restricting the use of agave species that are at risk may not be an adequate method for the conservation of some species, except in the most critically endangered cases, given that many Mexicans depend on their production to survive [2] and such restrictions might trigger a wave of poaching plants from wild populations. The great potential of sustainable management practices to ensure the future of particular agave species such as the maguey papalote (*A. cupreata*) and its derivatives and products has already been demonstrated. By providing greater visibility to sustainable practices by the local farmers, knowledge exchange between producers and science and preserving ancestral traditional production practices, agaves clearly have a profitable, sustainable, environment-friendly, socially responsible, and economically viable future [101,102].

It is also important to mention that the taxonomy of the genus *Agave* has been a complicated task for scientists, given that the genus is one of the most species-rich in the plant kingdom, and therefore recent discoveries that indicate synonymies and cryptic species are constant events, resulting in ongoing changes and descriptions of new species (see [31]). We must emphasize that there is still a great amount of data to collect on recently described species along with their distributions, natural history, current uses and pollinators to fully understand their biology, potential as providers of services and products, and conservation needs [31]. This study is just the beginning of an effort to update the knowledge about the uses, information available, and conservation needs of *Agave* species, and we hope to continue to facilitate updates in the future.

Agaves are among the most ecologically and economically important plants of Mexico. From an economic perspective, the role of *Agave* in production of distilled beverages and fibers, their ornamental relevance, and rural uses as living fences and building materials are very important for the country. In particular, exports of tequila and mezcal are undergoing a huge upward trend. Tequila exports went from 209 million liters in 2018 [103] to 249.4 million liters in 2021 with a net value of $2.3 billion dollars [17]. Mezcal exports went from 2.7 million liters to 7.1 million liters only from 2016 to 2018, with a value of $53 million dollars [103,104,105]. People directly or indirectly involved in the production of mezcal and tequila depend on the income generated by these distilled beverages. Additionally, Mexicans throughout the country depend on agaves for different products such as handcrafts, baskets or ropes made from their fibers, their use in Mexican cuisine for many typical dishes, and the preparation of the pre-Hispanic fermented drink, pulque.

Agaves are emblematic plants for Mexico and have been represented in codex, archeological vestiges and cultural traditions since pre-Hispanic times, being nowadays an extensive source of income and everyday life uses. We must prevent the extinction of agave species due to greedy, industrialized production and cultivation practices. Respect for traditional procedures, clear recognition and incorporation of traditional indigenous knowledge and preserving, creating, and implementing the best sustainable production practices is absolutely crucial for the future of agaves and their pollinators. Local communities demand this with reason, Mexico deserves this, and the world should expect nothing less. The future of *Agave* and its pollinators should be one of appreciation, respect, and responsible consumption of agave products, solidly anchored in environmental sustainability, social justice, and economic viability.

## 4. Conclusions

Our analysis highlights the great diversity of species of the genus *Agave* in Mexico, the intense level of use that some species are undergoing, the species-specific extinction risk level, and the lack of knowledge about most species. In the process of constructing this diagnostic analysis, it was clear that most of the available information for most species is scarce given their usefulness and distribution.

For the agaves used in mezcal production, we believe that the most convenient solution is to opt for strict sustainable management practices and favorable reproduction methods, while avoiding monoculture procedures that could affect genetic variability, and reduce or avoid the use of pesticides that affects pollinators thus affecting gene flow and connectivity between species. It is important to favor the natural sexual reproduction of the species, promoting natural pollination and protecting their pollinators. Traditional practices should be encouraged for the management of agaves, recognizing monocultures and high industrialization processes as inadequate and only happening under highly controlled conditions, ensuring the future of biodiversity and emphasizing the knowledge of local, traditional producers.

## Figures and Tables

**Figure 1 plants-12-00124-f001:**
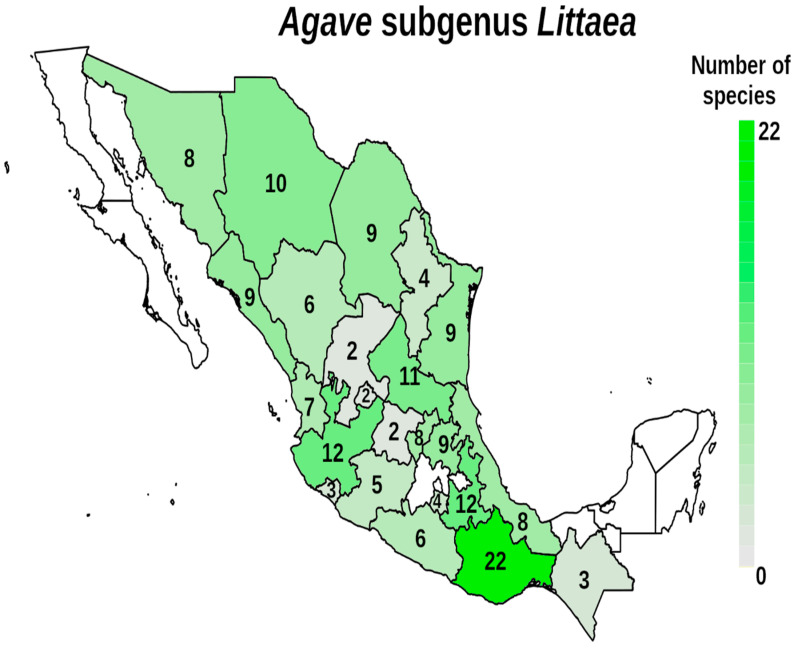
Distribution and number of species of the *Agave* subgenus *Littaea* by Mexican states. Color intensity is proportional to the number or species.

**Figure 2 plants-12-00124-f002:**
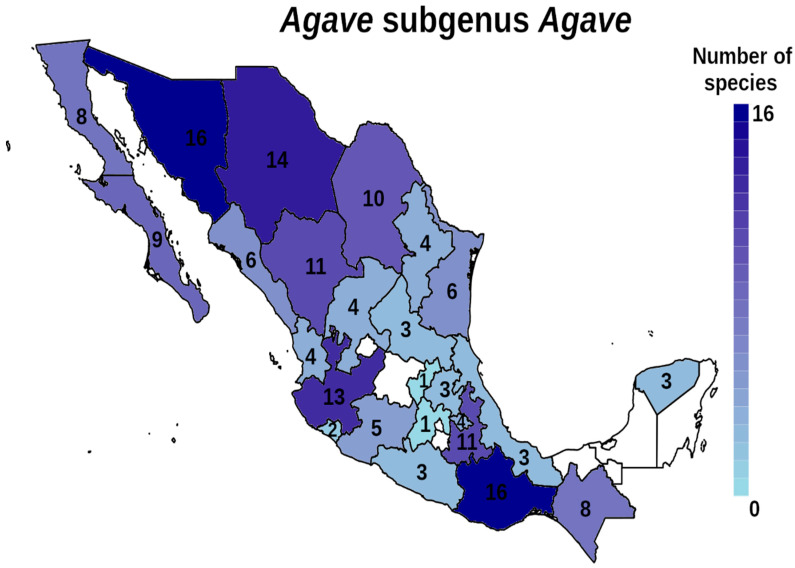
Distribution and number of species of the *Agave* subgenus *Agave* by Mexican states. Color intensity is proportional to the number or species.

**Figure 3 plants-12-00124-f003:**
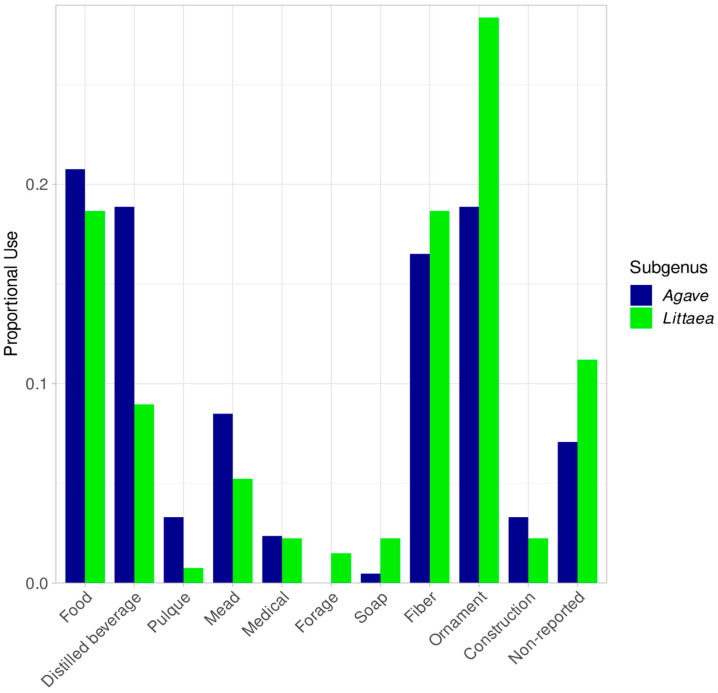
Proportional uses for *Littaea* and *Agave* subgenus.

**Figure 4 plants-12-00124-f004:**
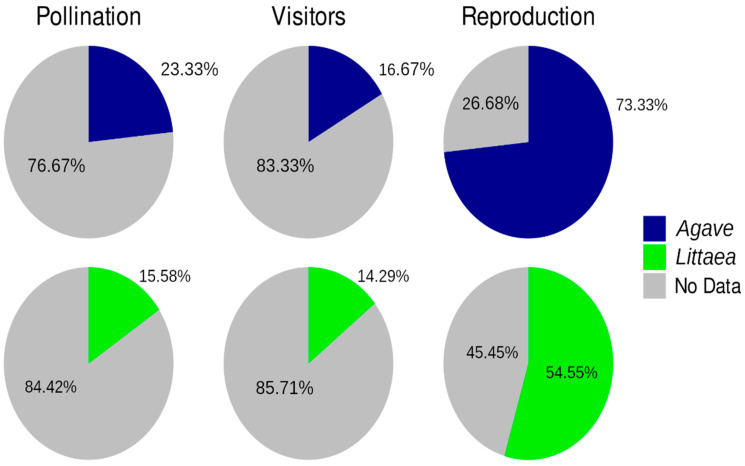
Proportion of the information available concerning pollination, visitors and reproduction for *Littaea* and *Agave* subgenus. Grey areas represent the absence of information.

**Figure 5 plants-12-00124-f005:**
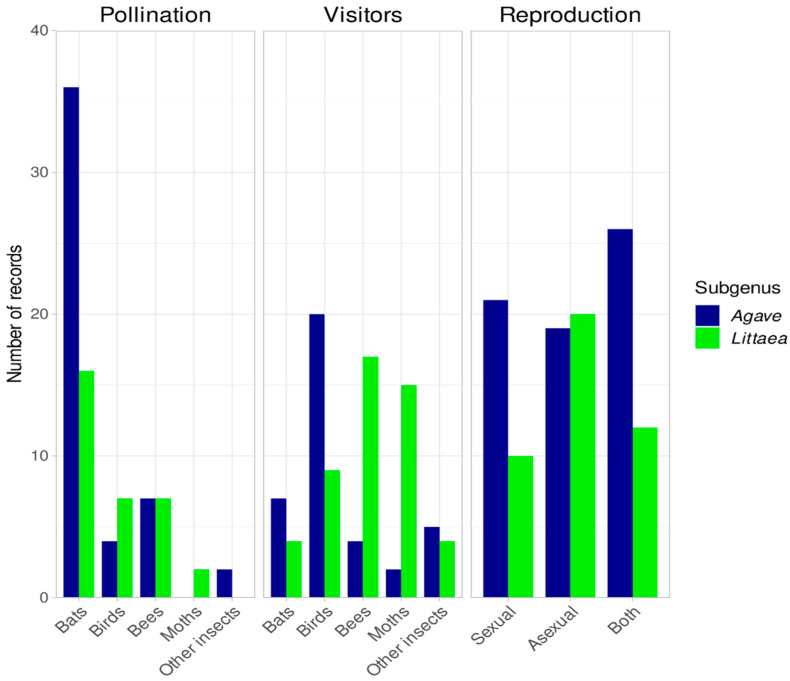
Number of records of the reproductive features analyzed for the *Littaea* and *Agave* subgenus. Pollinators were grouped in the following categories: Bats, Birds, Bees, Moths and Other insects; see Table 1 and Table 2 for details.

**Figure 6 plants-12-00124-f006:**
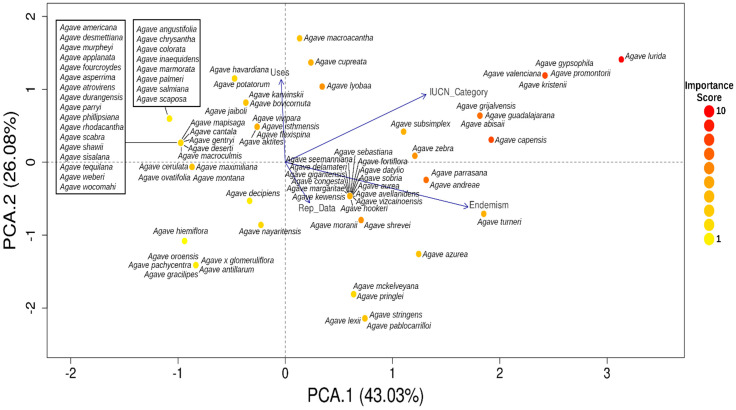
Principal Component Analysis of the *Agave* subgenus *Agave* using the IUCN category, uses, endemism, and reproductive data as variables (see Table 1 for details). Dot color represents the Importance Score obtained.

**Figure 7 plants-12-00124-f007:**
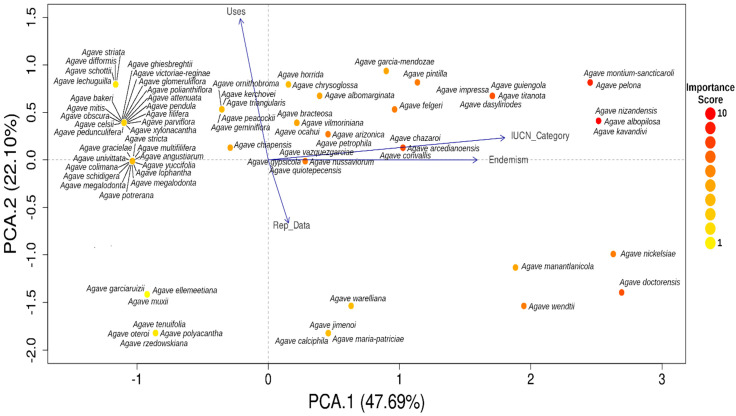
Principal Component Analysis of the *Agave* subgenus *Littaea* using the IUCN category, uses, endemism, and reproductive data as variables (see Table 2 for details). Dot color represents the Importance Score obtained.

**Table 3 plants-12-00124-t003:** Weighted score criteria regarding the data collected from the Agave species.

CR: 3 points EN: 2 points VU: 1 point EW: special case (4)	E: 2 points NE: 0 points	Has uses: 2 points Zero uses: 0 points	Knowledge of pollinator, floral visitor and reproduction: 0 points Knowledge of two or one aspects: 1 point No knowledge at all: 2 points

**Table 4 plants-12-00124-t004:** Species of subgenus Agave and Littaea under the established categories subjected to their conservation status and biological environment.

Subgenus *Agave*	Subgenus *Littaea*
Severely endangered: facing possible extinction; avoid use of wild plants	
*Agave lurida*	*Agave nizandensis*
**High risk of extinction: minimize use**	
*Agave capensis*	*Agave albopilosa*
*Agave gypsophila*	*Agave chiapensis*
*Agave kristenii*	*Agave dasyliriodes*
*Agave parrasana*	*Agave guiengola*
*Agave promontorii*	*Agave impressa*
*Agave valenciana*	*Agave kavandivi*
	*Agave montium-sancticaroli*
	*Agave pelona*
	*Agave titanota*
	*Agave pintilla (complex)*
**Threatened: ensure and prepare a management and recovery plan**	
*Agave abisaii*	*Agave arcedianoensis*
*Agave andreae*	*Agave bracteosa*
*Agave congesta*	*Agave chazaroi*
*Agave grijalvensis*	*Agave convallis*
*Agave guadalajarana*	*Agave doctorensis*
*Agave kewensis*	*Agave felgeri*
*Agave moranii*	*Agave gypsicola*
*Agave shrevei*	*Agave nussaviorum*
*Agave vizcainoensis*	*Agave petrophila*
*Agave zebra*	*Agave quiotepecensis*
	*Agave vazquezgarciae*
	*Agave victoria-reginae (complex)*
	*Agave nickelsiae (complex)*
	*Agave wendtii*

## Data Availability

Data are available in the published literature as discussed in the article.

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
