# Peer review of "Uses, Knowledge and Extinction Risk Faced by Agave Species in Mexico"

_plants, 2022, doi:10.3390/plants12010124_

Round 1

Reviewer 1 Report (Previous Reviewer 1)

I recall having received this manuscript for review before, but it was rejected at the time.

I see a significant improvement in quality - the Authors added charts and drawings that significantly improve the reception of the presented content.

Please provide a proof of language proofreading.

The goal and assumptions were clearly defined. This article provides comprehensive information. As many as 103 literature items were cited.

I only have to pay attention to the appearance of tables, drawings and charts - please refine them in terms of editing (use a uniform font, style and size, choose vivid colors, adjust the size of the charts to the layout of the article so that it constitutes a uniform whole).

I recommend for publication.

Author Response

Thank you for your encouragement and suggestions. We have had the entire manuscript read by three other people including one native English speaker and two with extensive experience publishing, writing, and speaking English. We are fully confident that this issue has been completely resolved.

We have also reformatted the tables and figures to be more reader-friendly, with better choice of colors and fonts. Thank you for your recommendation to publish. 

Reviewer 2 Report (Previous Reviewer 3)

I think this paper is ready for publication. The authors have made the minor revisions of the text required

Author Response

Thank you very much for your recommendation. We have had the manuscript read by three additional people, all very highly proficient in English and the issue has been resolved.

This manuscript is a resubmission of an earlier submission. The following is a list of the peer review reports and author responses from that submission.

Round 1

Reviewer 1 Report

The manuscript does not match the mdpi format (table, references, etc.).

The content of the article is very poor (especially the discussion), the introduction is too long in my opinion. The tables make up most of the content presented, but it is not innovative enough to be published in Plants.

Reviewer 2 Report

Comments for the Author:

Review on” We drink to your health: Uses, knowledge and extinction risk faced by Agave species in México” provide us with results of updated database analysis of the Agave species. Generally, I think that the topic of the paper is very interesting and brings new original data for this topic. The title is very promising, but unfortunately the contents of the article are very stingy. Above all, I lack more information about knowledge. If author do more analysis and expend discussion, I recommend a major revision.

Further comments:

Title:

 I think that “We drink to your health..” it is unnecessary and does not give us any key information.  

Keywords:

Delete keywords that are already in the title. It's a duplication.

Introduction:

It is too long.

Methods:

I miss distribution of the species on the map. I also miss citation/explanation how and from where author get the data. It is written in table this and this, but in the text must be summarised.

Line: 175: “An updated review of the Agave species ..” Which updated review?

Results and discussion:

Considering that the authors have a lot of data, I expected more in-depth analyses.

Reference: Reference list it is too long, it is really all necessary?

Table 3: There are no explanation of the abbreviations.

Reviewer 3 Report

this is a nice review covering all species in Mexico. I have not find any mistakes or shortcomings in the text. However, you do not need to repeat Agaves as a keyword, since they are already mentioned in title of the article. Table 1 is important and seems to be adequate. I have not checked all the species mentioned, but naming authors are given to all species, which is important in a publication like this.